# Molecular surveillance for *Rickettsia* spp. and *Bartonella* spp. in ticks from Northern Iran

**Ahmad Ghasemi**[1,2‡], **Mina Latifian**[2,3‡], **Saber Esmaeili**[2,3], **Saied Reza Naddaf**[4], **Ehsan Mostafavi**[2,3]*

**1** Department of Microbiology, Research Center of Reference Health Laboratories, Ministry of Health and Medical Education, Tehran, Iran, **2** National Reference Laboratory for Plague, Tularemia and Q Fever, Research Centre for Emerging and Reemerging Infectious Diseases, Pasteur Institute of Iran, Akanlu, KabudarAhang, Hamadan, Iran, **3** Department of Epidemiology and Biostatics, Research Centre for Emerging and Reemerging Infectious Diseases, Pasteur Institute of Iran, Tehran, Iran, **4** Department of Parasitology, Pasteur Institute of Iran, Tehran, Iran

‡ AG and ML contributed equally to this project and should be considered co-first authors.
* mostafaviehsan@gmail.com

**Data Availability Statement:** All relevant data are within the paper.

**Funding:** The study was supported by a grant from the National Institute for Medical Research Development (NIMAD) (grant no: 987978). Also

## Abstract

Tick-borne zoonotic diseases pose a threat to public health; hence, identifying the pathogenic agents associated with them is critical. The prevalence of *Bartonella* and *Rickettsia* in Iran is unknown. This study aimed to detect *Rickettsia* spp. and *Bartonella* species in ticks in northeast Iran and conduct phylogenetic analysis on these bacteria. Ticks from the sample bank in the Research Center for Emerging and Re-emerging Diseases were included in this study. The ticks were collected in 2017 and 2018 from domestic animals (sheep, goats, cows, camels, horses, dogs, and donkeys) and rodents in Golestan, Mazandaran, and Guilan provinces. Molecular methods were used to examine the DNA extracted from these samples to detect *Rickettsia* spp. and *Bartonella* species. The study examined a total of 3999 ticks. *Ixodes ricinus* (46.4%), *Rhipicephalus turanicus* (26.3%), and *Rhipicephalus sanguineus* (17.1%) were the most prevalent species. Among 638 DNA pools, real-time-PCR detected *Rickettsia* spp. in 161 (25.2%), mostly belonging to *Rh. sanguineus* (48.9%) and *Rh. turanicus* (41.9%). Golestan Province had the highest number of positive pools (29.7%). No positive samples for *Bartonella* were detected in a 638 pooled samples. Eight distinct *Rickettsia* species were detected in 65 sequenced samples, the majority of which were *R. massiliae* (n = 32, 49.2%) and *R. sibirica* (n = 20, 30.8%). Other species included *R. rhipicephali* (n = 3), *R. aeschlimannii* (n = 5), *R. helvetica* (n = 5), *R. asiatica* (n = 4), *R. monacensis* (n = 6), and *R. raoultii* (n = 1). The research findings may provide helpful information about tick-borne *Rickettsiae* in Iran and help to clarify the role of these arthropods in maintaining these agents. *Rickettsia* species were found to be circulating in three Northern provinces; thus, it is recommended that this disease be considered in the differential diagnosis of febrile diseases caused by tick bites and febrile diseases with skin rashes such as Crimean–Congo hemorrhagic fever (CCHF).

supported by Pasteur Institute of Iran (grant no: 1744) and Prof. Mostafavi received the fund.

**Competing interests:** The authors have declared that no competing interests exist.

# Introduction

Zoonotic diseases are a significant cause of infection-related mortality on a global scale and are critical as emerging infectious diseases in humans [1]. These diseases are a significant public health problem and a direct threat to human health, posing a risk of death. Climate change, travel and tourism, migration, animal trade, human factors, and natural factors significantly impact the spread of emerging and re-emerging infections [2].

Ticks are more likely than other blood-feeding arthropods to transmit pathogenic species, affecting humans and livestock [3]. Knowledge of the intricate relationship between ticks and their hosts, the environment, and the pathogens they transmit, is critical to understanding the epidemiology of tick-borne zoonotic diseases [4]. Ticks transmit various zoonotic diseases, including Lyme disease, Ehrlichiosis, and Rocky Mountain Spotted Fever (RMSF) [5]. Tick-borne diseases are a frequent occurrence in both human and veterinary medicine. The number of tick-borne diseases affecting domestic animals and humans has increased in recent years, necessitating additional research into the epidemiology, diagnosis, and ecology of these newly identified diseases [6].

*Rickettsia* infections transmitted by ticks are known as emerging and zoonotic diseases that affect human and livestock health. The distribution of these diseases worldwide is related to the vector [7]. *Rickettsia*, an obligate intracellular gram-negative bacterium, is an ancient vector-borne disease. This bacterium is transmitted via arthropods' bites to animals and humans. Ticks are the primary vectors for this pathogen; however, other vectors, including fleas, lice, mites, and even mosquitoes, may contribute to transmission. These bacteria are classified into more than 30 species, comprising four subgenera: spotted fever, typhus fever, *Rickettsia bellii*, and *Rickettsia canadensis*. The typhus group comprises *Rickettsia prowazekii* and *Rickettsia typhi*, while the spotted fever group with more than 20 species [8, 9]. Spotted fever group (SFG) is subdivided into two distinct subgroups: Rocky Mountain Spotted Fever (RMSF) and Mediterranean Spotted Fever (MSF). The Mediterranean spotted fever is an endemic disease in the Mediterranean region (Africa, Southern Europe, and India) and is believed to be associated with global warming and increased tick infestation and bites [10]. Rickettsial infections are transmitted to animals via tick bites, and if humans are in the vicinity of tick-infected animals, be bitten by ticks accidentally, are unable to propagate the infection in nature, and are known as dead end host for this pathogen [11].

In humans, the symptoms of this disease are nonspecific and are not diagnosed in time; as a result, necessary treatments are not performed in the majority of cases. Early symptoms include fever, chills, headache, and myalgia, and additional symptoms such as nausea, vomiting, and diarrhea can be seen in the subsequent steps. In untreated patients, mortality rates ranged from 5% to 25% [7]. Eight tick-borne rickettsios including: *Rickettsia rickettsii*, *Rickettsia sibirica*, *Rickettsia australis*, *Rickettsia honei*, *Rickettsia japonica*, *Rickettsia africae*, *Rickettsia conorii*, and *Rickettsia slovaca* have been described in a worldwide In recent years, human infections caused by *Rickettsia aeschlimannii*, *Rickettsia helvetica*, *Rickettsia mongolotimonae*, and *Rickettsia heilongjiangii* have also been reported in various geographical regions [12].

*Bartonella* is a vector-borne bacterium linked to increased emerging zoonotic infections in humans and animals. The infections range from mild or self-limiting to severe and life-threatening diseases in human. The bacterium is widespread among small mammals and may threaten human health [13]. *Bartonella* spp. are gram-negative, slow-growing, intracellular pleomorphic bacillus and includes at least 35 species and three subspecies. Numerous animals are considered hosts and reservoirs for this disease, primarily transmitted through flea and lice feces, sandflies, and possibly tick bites [14]. These bacteria cause chronic infections by infecting red blood cells and attacking endothelial cells, CD34$^+$ progenitor cells, and host dendritic cells [15].

*Bartonella* infects humans and a wide variety of animal species. *Bartonella* DNA has been detected in ticks suspected of being a source of animal-associated *Bartonella* infection in humans. *Bartonella* spp. cause a variety of diseases in humans, including cat-scratch disease (*Bartonella henselae*), Carrion's disease (CD) (*Bartonella bacilliformis*), trench fever (*Bartonella quintana*), endocarditis (*B. quintana* and *B. henselae*), bacillary angiomatosis (*B. quintana* and *B. henselae*), and hepatic peliosis (*B. henselae*) [5, 16]. *B. quintana* has emerged in homeless individuals who were in poor health [14]. The zoonotic pathogen *B. henselae*, which causes cat scratch disease, is probably the most prevalent in developed countries and is strongly associated with other syndromes, particularly eye infections and endocarditis. Cats are the main reservoirs; cat fleas (*Ctenocephalides felis*) are considered the primary vector, but other arthropods, e.g., ticks, may contribute to transmission [16].

Given the threat to public health posed by tick-borne zoonotic diseases, it is critical to study pathogens associated with ticks. There is limited information on the prevalence of *Bartonella* and *Rickettsia* in Iran and the prevalence of *Bartonella* and *Rickettsia* in human and vector infections. Thus, this study aimed to detect *Rickettsia* spp. and *Bartonella* species in ticks in northern Iran and conduct phylogenetic analysis.

## Methods

### Ethics statement

The Ethics Committee for National Institute for Medical Research Development "NIMAD" approved this study (Ethic Code: IR.NIMAD.REC.1398.373).

### Sample collection

Ticks from the Research Center for Emerging and Re-emerging Diseases' samples bank were used for analysis. The ticks were collected in 2017 and 2018 from domestic animals (sheep, goats, cows, camels, horses, dogs, and donkeys) and rodents in Golestan, Mazandaran, and Guilan provinces in northern Iran. The study area is bounded on the north by the Caspian Sea and the south by the Alborz mountain range. These provinces are covered with forests, mountains, and coasts and, have a temperate climate with three distinct geographical regions including temperate plains, mountainous and semi-arid regions.

Ticks samples were previously identified using morphological keys [17]. After the identification of collected ticks, the ticks were pooled for DNA extraction, based on the same tick's species, collected locations, hosts, tick sex, and the growth stage of ticks. Finally, the pools of ticks included 1 to 22 ticks based on the above criteria.

### DNA extraction from ticks

Pooled ticks were homogenized using liquid nitrogen and sterile PBS, and DNA was extracted using the potassium acetate method. Initially, 500 μl lysis buffer (0.1 M Tris-HCl, 0.05 M EDTA, 0.2 M sucrose, and 0.5% SDS) and 20 μl proteinase K (10 mg/ml) were added to the homogenized samples, and incubated at 56°C for one day. Amounts of 120 μl of 5 M sodium acetate were added to the samples and kept on ice for 10 min. After centrifuging the samples at 9660 RCF for 10 minutes, the supernatant was recovered, and 35 μl of 4 M sodium acetate and 1 ml of pure ethanol were added. The samples were thoroughly mixed and kept on ice for 10 minutes. The supernatant was discarded after centrifuging at 9660 RCF for 20 min, and the precipitates were washed with 500 μl of 70% ethanol, and the remaining alcohol was allowed to dry completely at room temperature. Finally, 200 μl of elution buffer (1 M Tris-HCl, 1 M EDTA) was added to the completely dried precipitates and kept at -20°C until the test [18].

**Table 1. The primers and the probe used for the detection of *Rickettsia* spp. & *Bartonella* spp.**

| Genus | Gene target | Sequence (5′ to 3′) | Amplicon size (bp) |
|---|---|---|---|
| *Rickettsia* spp. | Rsp | 5′- CGCAACCCTYATTCTTATTTGC -3′ | 149 |
| | | 5′- CCTCTGTAAACACCATTGTAGCA -3′ | |
| | | 6- FAM-TAAGAAAACTGCCGGTGATAAGCCGGAG-TAMRA | |
| *Bartonella* spp. | 16S-23S rRNA | 5′-GGGGAAGGTTTTCCGGTTTATC-3′ | 92 |
| | | 5′-GAGGACTTGAACCTCCGACC-3′ | |

## Detection of the *Rickettsia* genus

Using the TaqMan Real-Time PCR method, DNA extracted from ticks was analyzed for *Rickettsia* spp. The 20 μl reactions contained 10 μl commercial master mix (RealQ Plus 2x Master Mix Ampliqon, Denmark), 4 μl template DNA, 900 nmol of forward and reverse primers, 200 nmol of a probe (marked with 6-Carboxyfluorescein (6-Fam) fluorescent dye as a reporter dye and TAMRA as a quencher) [19], and sterile distilled water to final volume. *R. conorii* DNA (Amplirun, Vircell) and distilled water were included in all assays as positive and negative controls, respectively. The amplification was performed in a Corbett 6000 Rotor-Gene system (Corbett, Victoria, Australia) programmed for a 10-min activation at 95˚C, followed by 45 cycles at 95˚C for 15 sec, and 60˚C for 60 sec. The quantitative analysis was performed using the Rotor-Gene Q Series software and reading was taken at the end of each cycle in green at 60˚C (Table 1).

## Amplification of *gltA* gene

According to the result of Real-Time PCR, positive sample for *Rickettsia spp*. with a CT $\leq$ 30 were further analyzed to identify the *Rickettsia* species by amplifying the *gltA* gene. The 20 μl reactions contained 10 μl of 2X commercial master mix (2x TEMPase Master Mix RED A Ampliqon, Denmark), 4 μl of DNA template, 500 nmol of g1tA-F primer (GCTCTTCTCA TCCTATGGCTATTAT), and 500 nmol of g1tA-R primer (CAGGGTCTTCRTGCATTTCTT), and distilled water to the final volume [20]. The amplification program included an initial denaturation at 95˚C followed by 40 cycles at 94˚C for 30 sec, 58˚C for 30 sec, 72˚C for 60 sec, and final annealing at 72˚C for 7 min.

## Phylogenetic analysis

PCR products were electrophoresed on 1% agarose gel, and samples with specific bands (834 bp) were sent for sequencing (Genomin Co., Tehran, Iran). Chromas 2.6.6 software was used to evaluate the sequences. Final nucleotide sequences were compared with those available in the GenBank® database using the Basic Local Alignment Search Tool (http://blast.ncbi.nlm.nih.gov/Blast.cgi). The *gltA* gene sequences for various *Rickettsia* species were extracted from the GenBank database and phylogenetically analyzed using MEGA software in conjunction with the sequences obtained from the samples in this study (MEGA version X 10.1). Phylogenetic relationships were inferred using the Neighbor-Joining method based on the Kimura 2-parameter model for Rickettsia spp. Gamma distribution (+G) was used to model evolutionary rate differences among sites selected by the best-fit model. Evolutionary analyses were conducted on 1000 bootstrap replications using the MEGA X software.

**Bartonella detection.** The 16S-23S rRNA gene was targeted in ticks' DNA using the SYBR Green Real-time PCR method. The 20 μl reactions contained 10 μl of Plus 2x Master Mix Green Low ROX[TM] (AmpliQon, Denmark), 700 nmol of the forward primer and reverse primers (Table 1), 4 μl of template DNA, and distilled water to the final volume. The

amplification was programmed in a Corbett 6000 Rotor-Gene system (Corbett Victoria, Australia) for an initial denaturation at 95˚C for 10 min, followed by 40 cycles of 95˚C for 15 sec, 60˚C for 20 sec, and 72˚C for 20 sec with a melting step in between. *Bartonella henselae* DNA (Amplirun, Vircell) and distilled water were included in all assays as positive and negative control.

## Results

In this study, 3999 ticks were examined by molecular method. After morphological examinations, ticks were pooled according to species, location, sex, and growth stage. There were 1415 male ticks (35.38%), 2542 female ticks (63.56%), and 42 nymphs (1.05%). A total of 1926 ticks were collected from cattle (48.16%), 1458 ticks were collected from sheep (36.45%), 383 ticks were collected from goats (9.57%), 50 ticks were collected from dogs (1.25%), 128 ticks were collected from camels (3.20%), five ticks were collected from horses (0.12%), two ticks were collected from donkeys (0.05%), and 47 ticks were collected from hedgehogs (1.17%). The study identified 11 tick species using morphological keys. They were classified into four genera: *Ixodes*, *Haemaphysalis*, *Hyalomma*, and *Rhipicephalu*s, with the highest percentage belonging to *Ixodes ricinus* (46.4%), followed by *Rhipicephalus turanicus* (26.3%) and *Rhipicephalus sanguineus* (17.1%) (Table 2). The most numerous species in the prepared pools were *I. ricinus* (n = 259, 40.59%), *Rh. sanguineus* (n = 145, 22.73%), and *Rh. turanicus* (n = 131, 20.53%), respectively (Table 2).

### *Rickettsia* detection by Real-Time PCR

Of 638 DNA pooled samples tested by the Real-Time PCR, 161 pools (25.2%) were positive Rickettsia. *Rhipicephalus* genus had the highest percentage of positive pools (42.3%), and also, the highest percentage of positive pools belongs to *Rh. sanguineus* (48.9%), *Rh. turanicus* (41.9%), *Hyalomma marginatum* (29.5%), and *Haemaphysalis inermis* (25%) respectively (Table 2). *Rickettsia* infection was highest (42.3%) in the *Rhipicephalus* genus and lowest (5%)

**Table 2. The population of ticks collected in the studied provinces, and the prevalence of positive tick pools for *Rickettsia* spp.es in 2017–18.**

| Genus | Species | No. of collected ticks in each province | | | Total N (%) | Number of tested pools | No. of positive pools for Rickettsia sp. (%) | |
|---|---|---|---|---|---|---|---|---|
| | | Mazandaran N (%) | Golestan N (%) | Guilan N (%) | | | Per tick species | Per tick genus (%) |
| *Ixodes* | I. ricinus | 1805(76.6) | 6(0.41) | 140(74.8) | 1951(46.4) | 259 | 13(5) | 13(5) |
| *Haemaphysalis* | H. concinna | 6(0.25) | - | 1(0.53) | 7(0.16) | 4 | 0(0) | 1(11.1) |
| | H. inermis | 3(0.12) | - | 1(0.53) | 4(0.09) | 4 | 1(25) | |
| | H. punctata | - | - | 1(0.53) | 1(0.02) | 1 | 0 (0) | |
| *Hyalomma* | H. anatolicum | - | 40(2.74) | - | 40(1.0) | 10 | 1(10) | 15(25.8) |
| | H. marginatum | 56(2/37) | 65(4.46) | 16(8.5) | 137(3.42) | 44 | 13(29.5) | |
| | H. dromedarii | - | 18(1.23) | - | 18(0.45) | 4 | 1(25) | |
| *Rhipicephalus* | Rh. bursa | 48(2.03) | 1(0.06) | - | 49(1.22) | 13 | 4(3.07) | 132(42.3) |
| | Rh. sanguineus | 343(14.5) | 343(23.5) | 1(0.53) | 687(17.1) | 145 | 71(48.9) | |
| | Rh. turanicus | 61(2.58) | 983(67.5) | 9(4.81) | 1053(26.3) | 131 | 55(41.9) | |
| | Rh. annulatus | 34(1.4) | - | 18(9.6) | 52(1.3) | 23 | 2(8.69) | |
| **Total** | | 2356(100) | 1456(100) | 187 (100) | 3999(100) | 638 | 161 (25.23) | 161 (25.23) |

In this study, 638 pools of ticks were prepared for DNA extraction and molecular analysis. Finally, 59.2%(n = 378), 31.6% (n = 202), and 9.09% (n = 58) of the prepared pools belonged to Mazandaran, Golestan, and Guilan provinces, respectively (Table 3).

**Table 3. Prevalence of *Rickettsia* spp. in ticks based on counties and provinces.**

| Counties | provinces | Number of pools tested | The number of positive pools for *Rickettsia* spp. per genus (%) N |
|---|---|---|---|
| Golestan | Aq Qala | 67 | 16(23.8) |
| | Bandar Torkaman | 56 | 11(19.6) |
| | Aliabad | 8 | 3(37.5) |
| | Gorgan | 39 | 19(48.7) |
| | Gomishan | 8 | 1(12.5) |
| | Aliabad-e-Katul | 23 | 10(43.4) |
| | Gonbad Kavus | 1 | 0(0) |
| | **Total** | 202 | 60(29.7) |
| Mazandaran | Nur | 62 | 17(27.4) |
| | Amol | 78 | 18(23) |
| | Babol | 45 | 13(28.8) |
| | Sari | 71 | 20(28.1) |
| | Qaemshahr | 55 | 22(40) |
| | Mahmudabad | 16 | 2(12.5) |
| | Savadkuh | 31 | 3(9.67) |
| | Unknown | 20 | 3(15) |
| | **Total** | 378 | 98(25.9) |
| Guilan | Talesh | 46 | 2(4.34) |
| | Masal | 5 | 1(20) |
| | Rudsar | 1 | 0(0) |
| | Lahijan | 6 | 0(0) |
| | **Total** | 58 | 3(5.17) |

in the *Ixodes* genus ($P<0.001$). According to the number of positive pools in each province, 29.7% of Golestan province, 25.9% of Mazandaran province, and 17.5% of Guilan province were positive (Table 3). *Rickettsial* infection was significantly higher in ticks from Mazandaran and Golestan provinces than in ticks from Guilan ($P<0.001$). There was no statistically significant difference in *Rickettsia* infection between ticks from Mazandaran and Golestan provinces ($P = 0.33$).

## Identification and phylogenetic analysis of *Rickettsia* species

A total of 65 pool samples positive for Rickettsia were selected for species identification in such a way that the selected samples included different tick species, and from all the studied counties and from different hosts. Also, the Load of Rickettsia DNA (CT $\leq$30) was considered in sample selection for the phylogeny survey. Based on results of sequences Blast in Gen-Bank® and phylogenetic analysis, eight distinct species were identified from a total of 65 sequenced *Rickettsia* gltA samples, the majority of which were *R. massiliae* (49.2%) (n = 32) and *R. sibirica* (30.8%) (n = 20). Other species of *Rickettsia* identified in present study included *R. rhipicephali* (n = 3), *R. aeschlimannii* (n = 5), *R. helvetica* (n = 5), *R. asiatica* (n = 4), *R. monacensis* (n = 6), and *R. raoultii* (n = 1) (Table 4 and Fig 1).

*R. massiliae* infection was detected in ticks (*Rh. turanicus* and *Rh. sanguineous*)from different hosts (cattle, sheep, goats, and dogs) in Mazandaran and Golestan provinces. This species was identified in all *Rh. turanicus* ticks collected in Golestan province. Additionally, *R. massiliae* infection was detected in *Rh. sanguineus* ticks isolated from dogs and sheep in Golestan province and ticks isolated from dogs, sheep, and goats in Mazandaran province. Thirty samples (99.75%) of *R. massiliae* detected in this study were identical in gltA gene sequence, and

**Table 4. Distribution of different species of *Rickettsia* spp. identified in tick species and provinces in this study.**

| Tick specie | Province | Host | No. of positive sample typed | *Rickettsia* spp (N) |
|---|---|---|---|---|
| *Rh. turanicus* | Golestan | Cattle | 1 | *R. massiliae (1)* |
| | | Dog | 3 | *R. massiliae (3)* |
| | | Goat | 2 | *R. massiliae (2)* |
| | | Sheep | 4 | *R. massiliae (4)* |
| | Mazandaran | Cattle | 1 | *R. sibirica (1)* |
| | | Dog | 4 | *R. sibirica (2), R. massiliae (2)* |
| | | Goat | 2 | *R. sibirica (1), R. massiliae (1)* |
| | | Sheep | 7 | *R. sibirica (4), R. massiliae (3), R. rhipicephali (1)* |
| *Rh. sanguineous* | Golestan | Goat | 1 | *R. massiliae (1)* |
| | | Sheep | 4 | *R. massiliae (4)* |
| | Mazandaran | Cattle | 2 | *R. sibirica (1), R. massiliae (1)* |
| | | Goat | 8 | *R. sibirica (2), R. massiliae (4), R. rhipicephali (2)* |
| | | Sheep | 13 | *R. sibirica (8), R. massiliae (6), R. aeschlimannii (1)* |
| *Rh. bursa* | Mazandaran | ND | 1 | *R. sibirica (1)* |
| *I. ricinus* | Guilan | Cattle | 1 | *R. monacensis (1)* |
| | Mazandaran | Cattle | 6 | *R. helvetica (5), R. asiatica (4), R. monacensis (5)* |
| *H. marginatum* | Guilan | Cattle | 1 | *R. aeschlimannii (1)* |
| | Mazandaran | Cattle | 1 | *R. aeschlimannii (1)* |
| | | Goat | 1 | *R. aeschlimannii (1)* |
| | | Sheep | 2 | *R. aeschlimannii (1), R. raoultii (1)* |
| Total | | | 65 | *R. sibirica (20), R. massiliae (32), R. rhipicephali (3), R. aeschlimannii (5), R. helvetica (5), R. asiatica (4), R. monacensis (6), R. raoultii (1)* |

matched 100% to the reference strain *R. massiliae* MTU5 (CP000683.1) in BLAST analysis. Two samples (R33 and R22a) were identical (100% and 99.87%, respectively) to the Genbank reference strain *R. massiliae* AZT80 (CP003319.1) and *R. massiliae* MTU5 (CP000683.1), respectively. No *R. massiliae* DNA was detected in *R. bursa*, *I. ricinus*, and *H. marginatum* ticks.

*Rh. sanguineus*, *Rh. turanicus*, and *Rh. bursa* ticks showed *R. sibirica* infection in Mazandaran province. According to gltA gene blast analysis in the Genbank, the majority (80%) of *R. sibirica* sequencesgenerated in the study were identical to *R. sibirica* strain (MF098405.1), and three *R. sibirica* identified (15%) also had 99.78% similarity with the above reference strain. Furthermore, a sample was matched 100% with another *R. sibirica* strain (acc. no. MF098404.1). There were no positive samples for *R. sibirica* in *I. ricinus* or *H. marginatum* ticks. Moreover, there were no positive samples for *R. sibirica* in ticks from Guilan and Golestan provinces.

Only *H. marginatum* and *Rh. sanguineus* ticks in the Mazandaran and Guilan provinces were infected with *R. aeschlimannii*, the most frequently isolated *Rickettsia* from *H. marginatum* ticks collected from cattle (Mazandaran, Guilan), sheep (Mazandaran), and goats (Mazandaran). Based on the gltA gene blast in the GenBank, two *R. aeschlimannii* detected in the present study (R6 and R21c) had a 99.88% similarity to a *R. aeschlimannii* strain (KY411135.1) belonging to the *H. marginatum*, and *Rh. sanguineus* ticks in Mazandaran province. Also three samples R31 (Mazandaran), R35 (Mazandaran), and R55 (Guilan) which were detected in *H. marginatum* had a 99.78% similarity to a *R. aeschlimannii* strain (KU961540.1), 99.78% similarity to *R. aeschlimannii* (HQ335513.1), and a 100% similarity to *R. aeschlimannii* (MH932014.1) in the GenBank for gltA gene sequence blast.

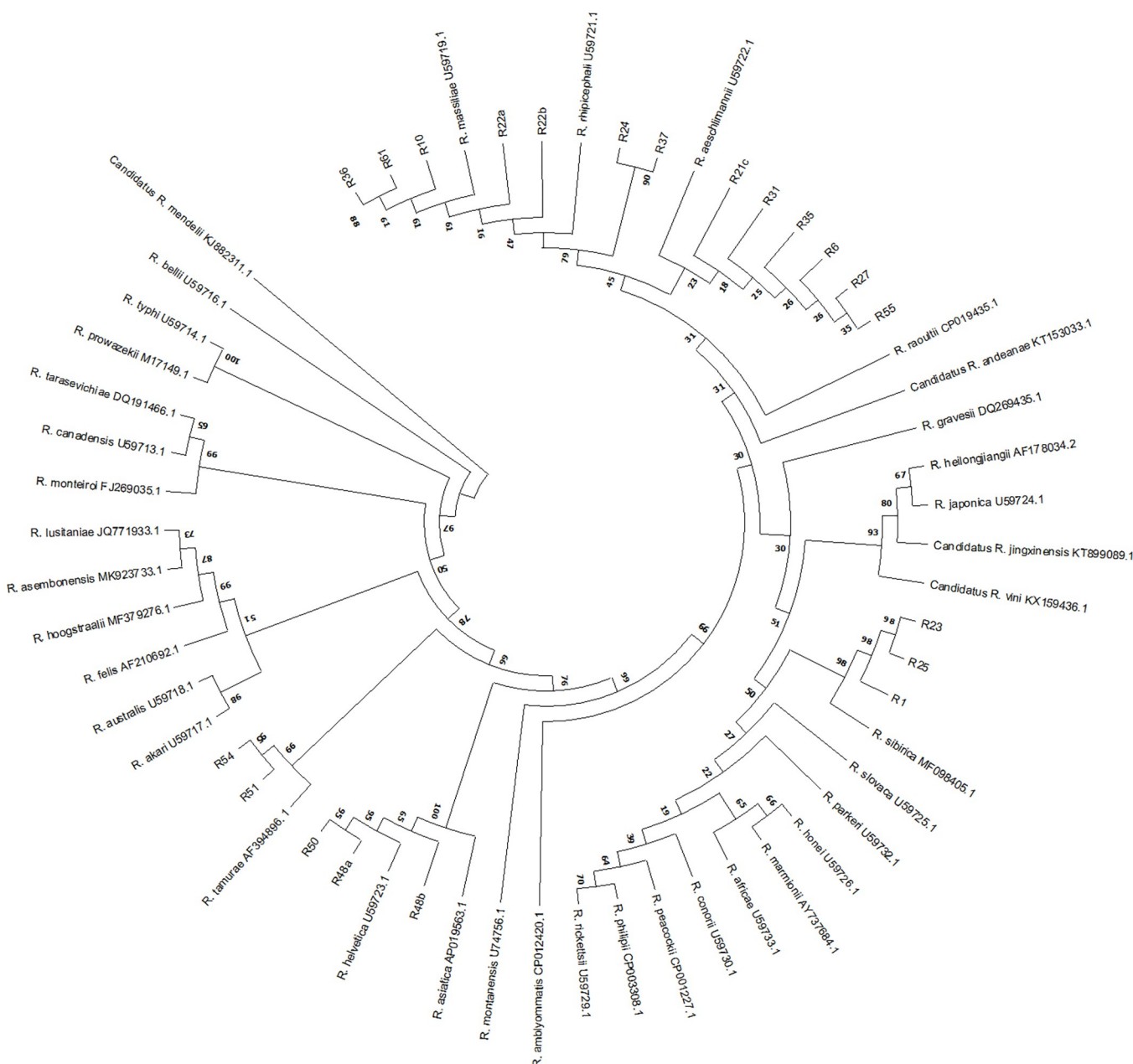

**Fig 1. Phylogenetic analysis based on *Rickettsia* gltA gene sequencing and Neighbor-Joining method algorithm (Kimura 2-parameter model).** The test was performed with bootstrap (1000 repetitions) by MEGA X 10.1 software. Sample ID included R_number were the studied samples in this study.

In Mazandaran province, *R. helvetica* was detected only in *I. ricinus* ticks isolated from cattle. Four *R. helvetica* samples (R48a, R49a, R52a, and R53a) were 100% identical to the *R. helvetica* sequence (KU310588.1) in the GenBank, and a sample (R50) was 99.88% identical to the recorded sequence in GanBanke database. Additionally, no infection with *R. helvetica* was detected in ticks collected from the other provinces.

*R. asiatica* infection was detected in four samples of *I. ricinus* isolated from cattle in Mazandaran province. When the gltA gene sequences of all samples were compared to the sequence of *R. asiatica* (AP019563.1) in the GenBank, 99.74% similarity was observed. Other tick species collected from other provinces were not infected with *R. asiatica*.

Furthermore, *R. monacensis* was isolated from *I. ricinus* ticks collected from cattle in Mazandaran and Guilan provinces. All five *R. monacensis* samples detected in the study were identical to the *R. monacensis* (LN794217.1) reference strains.

Only in Mazandaran province were *R. rhipicephali* infections observed in *Rh. sanguineus* (isolated from goats) and *Rh. turanicus* (isolated from sheep). When the gltA gene sequences of samples R22b, R24, and R37 were compared, they matched the *R. rhipicephali* (CP001313.1) reference strain in the GenBank by 97.87%, 99.5%, and 99.48%, respectively.

Moreover, we had a positive sample for *R. raoultii* in a *H. marginatum* tick isolated from sheep in Mazandaran province. The sample's gltA gene sequence was 99.83% identical to the sequence of *R. raoultii* (MT178338.1) in the GenBank.

Five samples contained three different *Rickettsia* species, four of which were *I. ricinus* isolates from cattle in Mazandaran province co-infected with *R. helvetica*, *R. asiatica*, and *R. monacensis*. Infections with *R. sibirica*, *R. massiliae*, and *R. aeschlimannii* were detected in a sample of *Rh. sanguineus* ticks isolated from sheep in Mazandaran province. Furthermore, an *Rh. turanicus* isolate from sheep in Mazandaran province was found to be co-infected with *R. massiliae* and *R. sibirica*.

### *Bartonella* detection

Based on the results of Real-time PCR, no positive samples for *Bartonella* spp. were detected among 638 tick pools.

### Discussion

Zoonotic diseases are a significant threat to public health. Here, we examined *Rickettsia* and *Bartonella* pathogens in ticks from three northern Iranian provinces using Real-time PCR to better understand their epidemiology and possible transmission via vectors. According to the results, *Rickettsia* spp. was detected in 161 of 638 pools (25.2%).

Rickettsial pathogens have received significant attention over the last 25 years, as several tick-borne species previously considered non-pathogenic are now associated with human infections [21]. There are few studies for detection of Rickettsia spp on ticks in Iran. However, numerous studies have been conducted in neighboring countries of Iran on this pathogen as an infectious and high-risk agent for public health [22]. According to the results of gltA gene sequencing on 65 positive samples, the majority of species were identified as belonging to *R. massiliae* (49.2%) and *R. sibirica* (30.8%). *R. massiliae* was initially isolated from the *Rhipicephalus* genus, and the majority of reports in France, Greece, and Spain have been associated with *Rh. turanicus* and *Rh. sanguineus* [23]. The first human infection with this species was reported in 2005 in Italy. In humans, *R. massiliae* causes a disease similar to MSF [24].

*R. massiliae* was detected in ticks *Rh. turanicus* and *Rh. sanguineus* isolated from cattle, sheep, goats, and dogs in the provinces of Golestan, Mazandaran, and Guilan. In Palestine, a study was conducted to detect Rickettsial SFG in 867 ticks collected from various hosts. In this study, 148 samples were identified as positive for *Rickettsia* infection *R. massiliae* was the most frequently identified *Rickettsia* in this study. Twenty-eight ticks were positive for this species, with the most positive samples belonging to *Rh. turanicus*. *Rh. sanguineus* ticks were collected from dogs and sheep, corroborating the current study [25].

In 1991, *R. sibirica* was isolated for the first time from *H. asiaticum* ticks in Mongolia. This pathogen was subsequently isolated from members of the genus *Hyalomma*, particularly *H. truncatum* and *H. excavatum*, worldwide. The first human infection with *R. sibirica* was reported in 1996 in France, and the patient presented with a mild illness that included fever, skin rashes, and scarring *R. sibirica* was identified in *Rh. sanguineus*, *Rh. turanicus*, and *Rh.*

*bursa*, but not in any of the *Hyalomma* genera in this study [26]. *R. aeschlimannii* was initially isolated in 1997 from *Hy. marginatum* ticks on cattle in Hungary. Furthermore, this species infects *H. marginatum* in Niger, Zimbabwe, and Mali. In humans, *R. aeschlimannii* causes a febrile illness similar to MSF, and the first case was diagnosed in a tourist to Hungary in 2000 [27]. As with the previous studies, most positive samples for *R. aeschlimannii* were detected in *H. marginatum* ticks isolated from cattle, sheep, and goats. Moreover, it was detected in the *Rh. sanguineus* ticks isolated from sheep in Mazandaran.

According to a study in Turkey, from 322 ticks were isolated from humans *Rickettsia* spp. was detected in 100 samples, and following species were identified: *R. aeschlimannii*, *R. slovaca*, *R. raoultii*, *R. hoogstraalii*, *R. sibirica*, and *R. monacensis*. *R. aeschlimannii* was the most frequently detected species (19.5%), in *Hy. marginatum* ticks. After *R. massiliae*, the most positive samples in present study belonged to *R. sibirica*, but in Turkey, only 0.31% of samples were infected with *R. sibrica* [28].

*R. monacensis*, a member of the spotted fever group, was isolated for the first time using molecular methods from *I. ricinus* ticks in Germany. It has also been isolated from *I. ricinus* ticks in Hungary and Portugal. The pathogenesis of *R. monacensis* is unknown, but human infections with this species have been reported in Spain, Italy, and the Netherlands [29]. In this study, *R. monacensis* was also detected in *I. ricinus* ticks in Mazandaran.

In a study conducted in Turkey, 25 out of 1019 ticks collected from various country regions were found to be positive for *Rickettsia* infection. Four of the 25 samples were reported positive for *R. monacensis*, only in *I. ricinus* ticks, and was consistent with the findings of this study. Additionally, *R. raoultii* was detected in *Dermacentor marginatus* ticks in the study in Turkey [30], but only positive samples for *R. raoultii* were detected in *H. marginatum* isolated from sheep in Mazandaran province. *R. raoultii* was first discovered in 1999 in *Rhipicephalus pomilius* and *Dermacentor nuttalli* ticks in the former Soviet Union and was later named *R. raoultii* in 2008 as a member of the *Rickettsia* family. Moreover, this pathogen was identified in *D. marginatum* isolated from a patient in France, demonstrating the pathogen's pathogenicity in humans [31].

*R. helvetica*, a member of the spotted fever group, was first detected in many European and Asian countries from *I. ricinus* ticks. Although this species is generally considered non-pathogenic, mild to severe disease cases, have been reported in humans [32]. *R. helvetica* was isolated from *I. ricinus* ticks collected from cattle in this study. in Turkey, 69 of 167 ticks isolated from humans hospitalized across the country tested positive for *Rickettsia* infection. According to the findings, this study found that *R. monacensis* (70%) was the most frequently detected species in *I. ricinus* ticks. Moreover, 2 (3%) *I. ricinus* ticks tested positive for *R. helvetica*, consistent with our findings in present study [33].

*R. rhipicephali* was isolated for the first time from *Rh. sanguineus* ticks in Mississippi but has not been associated with human infections [34]. *R. rhipicephali* was also found in *Rh. sanguineus* and *Rh. turanicus* ticks in this study. *R. asiatica*, originally designated *Rickettsia IO-1T*, was isolated from *Ixodes* ticks in Japan in 1993 [35]. *R. asiatica* was detected in *I. ricinus* ticks isolated from cattle in this study. The *R. rhipicephali* and *R. asiatica* species appeared to have not been isolated in neighboring countries and were described for the first time in this study.

In this study, the collected ticks were identified using only morphological keys. It is recommended to do the molecular identification of ticks as a confirmatory in future studies. Another limitation was that the main focus of the current study was on the detection of *Bartonella* spp that they are pathogenic in humans and animals. Therefore, it is possible that we could not detect some non-pathogenic *Bartonella*.

## Conclusion

According to the result of our study, *Rickettsia* species were found to be circulating in three Northern provinces. Based on data available in the world, a number of these identified pathogens in this study are capable of causing disease in humans. This is a cautionary tale for the health care system. As a result, it is recommended that these diseases be considered in the differential diagnosis of febrile diseases caused by tick bites, skin rashes, or Crimean–Congo hemorrhagic fever (CCHF) and that the healthcare system be educated about the disease's critical importance and symptoms.

## Acknowledgments

We would also express our gratitude to Dr. Ahmad Mahmoudi, Hamed Hanifi, Amir Hesam Neamati, Alireza Mordadi, Ali Mohammadi, and Seyyed Adel Hosseini from Department of Epidemiology and Biostatistics of Pasteur Institute of Iran, who helped us in tick's collection and some of the laboratory process.

## Author Contributions

**Conceptualization:** Ehsan Mostafavi.

**Data curation:** Ehsan Mostafavi.

**Funding acquisition:** Ehsan Mostafavi.

**Investigation:** Ahmad Ghasemi, Mina Latifian, Saber Esmaeili.

**Methodology:** Saber Esmaeili, Saied Reza Naddaf.

**Resources:** Saber Esmaeili, Saied Reza Naddaf.

**Writing – original draft:** Ahmad Ghasemi, Mina Latifian, Saber Esmaeili.

**Writing – review & editing:** Ahmad Ghasemi, Mina Latifian, Ehsan Mostafavi.

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
