## [Decision Letter · Decision Letter 0]

6 Oct 2022

PONE-D-22-25506Molecular Detection of Infection with Rickettsia spp. and Bartonella spp. in collected Ticks from Northern IranPLOS ONE

Dear Dr. Mostafavi,

Thank you for submitting your manuscript to PLOS ONE. After careful consideration, we feel that it has merit but does not fully meet PLOS ONE’s publication criteria as it currently stands. Therefore, we invite you to submit a revised version of the manuscript that addresses all of the points raised during the review process. In addition, I ask that you revise the title of your manuscript. The current title implies that you detected Bartonella in ticks, whereas you did not actually find any.  Perhaps something like:Molecular Surveillance for Rickettsia spp. and Bartonella spp. in Ticks from Northern Iran

We look forward to receiving your revised manuscript.

Kind regards,

Brian Stevenson, Ph.D.

Academic Editor

PLOS ONE

Reviewers' comments:

Reviewer's Responses to Questions

**Comments to the Author**

1. Is the manuscript technically sound, and do the data support the conclusions?

Reviewer #1: Yes

Reviewer #2: Partly

Reviewer #3: Yes

2. Has the statistical analysis been performed appropriately and rigorously? 

Reviewer #1: Yes

Reviewer #2: Yes

Reviewer #3: Yes

3. Have the authors made all data underlying the findings in their manuscript fully available?

Reviewer #1: Yes

Reviewer #2: Yes

Reviewer #3: Yes

4. Is the manuscript presented in an intelligible fashion and written in standard English?

Reviewer #1: Yes

Reviewer #2: Yes

Reviewer #3: Yes

5. Review Comments to the Author

Reviewer #1: The manuscript by Ghasemi et al., present data on Molecular Detection of Infection with Rickettsia spp. and Bartonella spp. in collected Ticks from Northern Iran. Overall, the manuscript is well described, and the information is interesting and important for researchers in the field of tick-borne diseases and ticks.

Only minor revisions are suggested to improve the manuscript.

Results section: the authors mentioned the identification of 11 collected tick species Line 197- using only morphological keys, some of them are very similar it would be good if they were able to do molecular identification of tick species as a confirmatory for morphological keys.

Table 1 title: Bartonella spp (spp. Shouldn’t be italic)

Table 3 title: Bartonella spp (spp. Shouldn’t be italic)

Line 180 table 1 remove reference 20 from the table title and if it is for the primers sets put in the method section.

Line 212: (P<0.001) P should be italic

Line 182 table 1 Bartonella correct to italic

Line 297 correct Bartonella to italic

Line 298 correct Bartonella to italic

Reviewer #2: Ghasemi et al. present an interesting piece of descriptive work on the prevalence of Rickettsia species in a region of Iran that includes ~7.5 million people that could be at risk of tick-borne illness. The work does a reasonable job cataloging Rickettsia species in the region, but I would like to see additional controls confirming their finding that Bartonella species were not present in ticks in the region. Otherwise, my concerns are largely minor and should not hold up publication.

Major Concerns

1) If the authors choose to include their Bartonella result, a positive control is necessary. The best experiment would be to amplify DNA from an arthropod known to be infected with Bartonella. The second best experiment would be Bartonella DNA spiked into one of their tick pools to confirm that the authors can detect Bartonella using their methods. qPCR on purified Bartonella DNA itself would not suffice.

2) Caution should be used comparing frequencies across the pools and a sentence pointing out that it is possible that the proportions of infections across species could have been skewed by the pool design (eg. Uneven distribution of infections across the pools) may be important to include around line 216 or so. Regardless, the findings themselves are interesting.

Minor Concerns

1) Some additional detail on pool design is necessary—is it correct to assume the pool size was around 5-6 ticks per pool? It appears that pools also contained a single species from a single region, is that correct? Details should be included in the methods, as well as when the study design is introduced around line 193.

2) Line 63 – I believe typhus fever, rather than Typhoid fever, is meant here.

3) Line 71 – Propagate/Spread may be more precise language than sustain here.

4) Line 228 – How were these 65 pools chosen. Table 4 suggests that the ticks and regions they came from were diverse, but some information in the text could be useful

5) Methods – Reporting centrifugation steps in terms of “times g” or “RCF” is preferred to RPM.

Reviewer #3: In this manuscript the authors have made a screening of the different rickettsia species present in northern Iranian regions. Using qPCR methodology they detected many positive ticks for rickettsia and were able to identify and classify the bacteria species.

The minor concerns I have focusses on the text, and some standard name usage.

1. It is complicated to differentiate Rickettsia (e.g R. sibrica) from Rhipicephalus ticks (R. sanguineus). I propose that authors change the writing of Rhipicephalus tick species in the whole manuscripts and use Rh. as done in the discussion (line 327).

Some part of text could be better explained for non-specific reader

line 70 It is hard to understand how animal transmission differ from human one, this part need better explanations.

line 122 I do not understand the meaning of "separated into groups of one to several" maybe you could said: After identifying tick samples, we separated ticks in groups of one to several individuals using different criteria: species, host, sex....

line 194 Could the authors give further details concerning the molecular analysis performed.

line 208 Rickettsia need to be added after the word "positive", to be sure that we understand the test conducted.

line 335 Re-write the "from 322 ticks were isolated"

line 346 wrong verb conjugation "In a study was conducted in Turkey" take out "was"

line 371-372 the author should explain explicitly what are the study limitations. These sentences are hard to understand.

Minor typos in the text:

line 72 missing space diagnosedin

line 117 is covered is use twice, may delete one.

line 133 . in a middle of a sentence (after the world ethanol)

line 249 In Cap i not necessary

line 256 sequneces to be change to sequences

line 268 Alsothree missing spaces between the 2 world

line 353 andwas missing space

6. PLOS authors have the option to publish the peer review history of their article (what does this mean?). If published, this will include your full peer review and any attached files.

Reviewer #1: No

Reviewer #2: **Yes: **Jeffrey Bourgeois

Reviewer #3: **Yes: **Marine J. Petit

---

## [Author Response · Author response to Decision Letter 0]

5 Nov 2022

Response to Editor and Reviewer Comments

From the handling editor:

Thank you for submitting your manuscript to PLOS ONE. After careful consideration, we feel that it has merit but does not fully meet PLOS ONE’s publication criteria as it currently stands. Therefore, we invite you to submit a revised version of the manuscript that addresses all of the points raised during the review process.

Editor Comments 1: In addition, I ask that you revise the title of your manuscript. The current title implies that you detected Bartonella in ticks, whereas you did not actually find any. Perhaps something like: Molecular Surveillance for Rickettsia spp. and Bartonella spp. in Ticks from Northern Iran

Author’s response 1: Thanks for your kind attention. The manuscript was revised based on editor comments.

Reviewer #1: 

The manuscript by Ghasemi et al., present data on Molecular Detection of Infection with Rickettsia spp. and Bartonella spp. in collected Ticks from Northern Iran. Overall, the manuscript is well described, and the information is interesting and important for researchers in the field of tick-borne diseases and ticks. 

Only minor revisions are suggested to improve the manuscript.

Reviewer Comments 1: Results section: the authors mentioned the identification of 11 collected tick species Line 197- using only morphological keys, some of them are very similar it would be good if they were able to do molecular identification of tick species as a confirmatory for morphological keys.

Author’s response 2: Thanks for your attention. In this study, the collected ticks were identified using only morphological keys. We understand your concern, but unfortunately, it is not possible for us to do it now, and we will try to consider your valuable advice in the future studies. Also, this limitation was added to the manuscript.

Reviewer Comments 2: Table 1 title: Bartonella spp (spp. Shouldn’t be italic)

Table 3 title: Bartonella spp (spp. Shouldn’t be italic)

Line 180 table 1 remove reference 20 from the table title and if it is for the primers sets put in the method section.

Line 212: (P<0.001) P should be italic

Line 182 table 1 Bartonella correct to italic

Line 297 correct Bartonella to italic

Line 298 correct Bartonella to italic

Author’s response 1: Thanks. The manuscript was revised based on the reviewer comments.

Reviewer #2: 

Ghasemi et al. present an interesting piece of descriptive work on the prevalence of Rickettsia species in a region of Iran that includes ~7.5 million people that could be at risk of tick-borne illness. The work does a reasonable job cataloging Rickettsia species in the region, but I would like to see additional controls confirming their finding that Bartonella species were not present in ticks in the region. Otherwise, my concerns are largely minor and should not hold up publication.

Major Concerns

Reviewer Comments 1: If the authors choose to include their Bartonella result, a positive control is necessary. The best experiment would be to amplify DNA from an arthropod known to be infected with Bartonella. The second best experiment would be Bartonella DNA spiked into one of their tick pools to confirm that the authors can detect Bartonella using their methods. qPCR on purified Bartonella DNA itself would not suffice.

Author’s response 1: We understand your concern. The negative result of Bartonella detection in all tick samples was surprising for us as well. We have checked our diagnosis protocol several times, and we tried different methods to make sure the results obtained for Bartonella are correct. Such as, before DNA extraction, we added a certain amount of positive control DNA to a few tick samples. Then the genomic DNA of these ticks was extracted and Bartonella was successfully detected in these samples by qPCR. Also, we spiked purified Bartonella DNA to a few extracted DNA of ticks, and Bartonella was successfully detected in these samples by qPCR, too.

Reviewer Comments 2: Caution should be used comparing frequencies across the pools and a sentence pointing out that it is possible that the proportions of infections across species could have been skewed by the pool design (eg. Uneven distribution of infections across the pools) may be important to include around line 216 or so. Regardless, the findings themselves are interesting.

Author’s response 2: Please see our previous response.

Reviewer Comments 3: Some additional detail on pool design is necessary—is it correct to assume the pool size was around 5-6 ticks per pool? It appears that pools also contained a single species from a single region, is that correct? Details should be included in the methods, as well as when the study design is introduced around line 193.

Author’s response 3: Thanks, the manuscript was revised.

“After the identification of collected ticks, the ticks were pooled for DNA extraction, based on the same tick’s species, collected locations, hosts, tick sex, and the growth stage of ticks. Finally, the pools of ticks included 1 to 22 ticks based on the above criteria”.

Reviewer Comments 4: Line 63 – I believe typhus fever, rather than Typhoid fever, is meant here. 

Author’s response 4: Thanks, revised.

Reviewer Comments 5: Line 71 – Propagate/Spread may be more precise language than sustain here.

Author’s response 5: Thanks, revised.

Reviewer Comments 6: Line 228 – How were these 65 pools chosen. Table 4 suggests that the ticks and regions they came from were diverse, but some information in the text could be useful 

Author’s response 6: A total of 65 positive Rickettsia samples were selected for the phylogenetic study based on the diversity of tick species, host, and geographical area. Also, the Load of Rickettsia DNA (CT ≤30) was considered in sample selection for the phylogeny survey.

 Reviewer Comments 7: Methods – Reporting centrifugation steps in terms of “times g” or “RCF” is preferred to RPM.

Author’s response 7: Thanks, revised.

Reviewer #3: 

In this manuscript the authors have made a screening of the different rickettsia species present in northern Iranian regions. Using qPCR methodology they detected many positive ticks for rickettsia and were able to identify and classify the bacteria species.

The minor concerns I have focusses on the text, and some standard name usage.

Reviewer Comments 1: It is complicated to differentiate Rickettsia (e.g R. sibrica) from Rhipicephalus ticks (R. sanguineus). I propose that authors change the writing of Rhipicephalus tick species in the whole manuscripts and use Rh. as done in the discussion (line 327).

Author’s response 1: Thanks, revised. 

Reviewer Comments 2: Some part of text could be better explained for non-specific reader:

line 70 It is hard to understand how animal transmission differ from human one, this part need better explanations.

line 122 I do not understand the meaning of "separated into groups of one to several" maybe you could said: After identifying tick samples, we separated ticks in groups of one to several individuals using different criteria: species, host, sex....

line 194 Could the authors give further details concerning the molecular analysis performed.

line 208 Rickettsia need to be added after the word "positive", to be sure that we understand the test conducted.

line 335 Re-write the "from 322 ticks were isolated":

line 346 wrong verb conjugation "In a study was conducted in Turkey" take out "was"

Author’s response 2: Thanks. All done.

Reviewer Comments 3: line 371-372 the author should explain explicitly what are the study limitations. These sentences are hard to understand.

Author’s response 3: The limitations of study were revised.

Reviewer Comments 4: Minor typos in the text:

line 72 missing space diagnosedin

line 117 is covered is use twice, may delete one.

line 133 . in a middle of a sentence (after the world ethanol)

line 249 In Cap i not necessary

line 256 sequneces to be change to sequences

line 268 Alsothree missing spaces between the 2 world

line 353 andwas missing space

Author’s response 4: Thanks for your kind attention. The manuscript was revised based on your comments

---

## [Editor Report · Decision Letter 1]

21 Nov 2022

Molecular Surveillance for Rickettsia spp. and Bartonella spp. in Ticks from Northern Iran

PONE-D-22-25506R1

Dear Dr. Mostafavi,

We’re pleased to inform you that your manuscript has been judged scientifically suitable for publication and will be formally accepted for publication once it meets all outstanding technical requirements.

Kind regards,

Brian Stevenson, Ph.D.

Academic Editor

PLOS ONE
---

## [Editor Report · Acceptance letter]

28 Nov 2022

PONE-D-22-25506R1 

Molecular Surveillance for *Rickettsia* spp. and *Bartonella* spp. in Ticks from Northern Iran 

Dear Dr. Mostafavi:

I'm pleased to inform you that your manuscript has been deemed suitable for publication in PLOS ONE. Congratulations! Your manuscript is now with our production department. 

Kind regards, 

on behalf of

Prof. Brian Stevenson 

Academic Editor

PLOS ONE